# Reducing Tyre Wear Emissions of Automated Articulated Vehicles through Trajectory Planning [note 1]

**DOI:** 10.3390/s24103179

**Published:** 2024-05-16

**Authors:** Georgios Papaioannou, Vallan Maroof, Jenny Jerrelind, Lars Drugge

**Affiliations:** 1Cognitive Robotics, TU Delft, 2628 CD Delft, The Netherlands; 2The Centre for ECO2 Vehicle Design, KTH Royal Institute of Technology, 100 44 Stockholm, Sweden; vallan@kth.se (V.M.); jennyj@kth.se (J.J.); larsd@kth.se (L.D.); 3Engineering Mechanics, KTH Royal Institute of Technology, 100 44 Stockholm, Sweden

**Keywords:** trajectory planning, optimal control, articulated vehicles, tyre wear, energy-efficiency

## Abstract

Effective emission control technologies and eco-friendly propulsion systems have been developed to decrease exhaust particle emissions. However, more work must be conducted on non-exhaust traffic-related sources such as tyre wear. The advent of automated vehicles (AVs) enables researchers and automotive manufacturers to consider ways to further decrease tyre wear, as vehicles will be controlled by the system rather than by the driver. In this direction, this work presents the formulation of an optimal control problem for the trajectory optimisation of automated articulated vehicles for tyre wear minimisation. The optimum velocity profile is sought for a predefined road path from a specific starting point to a final one to minimise tyre wear in fixed time cases. Specific boundaries and constraints are applied to the problem to ensure the vehicle’s stability and the feasibility of the solution. According to the results, a small increase in the journey time leads to a significant decrease in the mass loss due to tyre wear. The employment of articulated vehicles with low powertrain capabilities leads to greater tyre wear, while excessive increases in powertrain capabilities are not required. The conclusions pave the way for AV researchers and manufacturers to consider tyre wear in their control modules and come closer to the zero-emission goal.

## 1. Introduction

Electric and hybrid propulsion systems are expected to have increased non-exhaust pollution-related sources, such as tyre wear, due to their significantly increased mass [1,2]. As a result, the minimisation of tyre wear has attracted significant interest by academia and industry.

Tyre wear increases microplastics in the air, which greatly affects human health and environmental pollution [3,4,5]. According to a survey [6], around 18% of air pollution is because of exhaust emissions, while 11% is from tyre wear. At the same time, another survey estimated that the potential amount of tyre wear from heavy ground vehicles (HGVs) was around 2 million tonnes in 2013 among various countries in the world [7]. Therefore, it is evident that tyre wear is one of the most dominant non-exhaust pollution sources, and attempts to minimise it should be considered [8]. In this way, the potential benefits from electrification could be secured and the European Union’s (EU’s) goal for net-zero emissions could be achieved by 2050 [9].

Tyre wear occurs because of friction during the sliding between the tread surface and the road, while it is categorised as normal and abnormal [10]. Internal and external factors mainly affect tyre wear. The internal factors refer to the tyre shape, structure, material, and inflation pressure, while external factors refer to vehicle load, suspensions, driving dynamics and roads, etc. [11,12,13]. In HGVs, where this work focuses, scrubbing is generated by high lateral forces during cornering events, and this leads to extensive wear due to the high sideslip angles that occur [14].

At the same time, multi-axle and articulated vehicles generate higher lateral forces causing large abrasive wear, while unsteered axles have higher lateral forces due to the increased sideslip angle at these axles [15]. All these issues could be considered with the advent of automated vehicles (AVs). The megatrend of AVs is already extending to HGVs and articulated vehicles, with various companies around the world (such as Einride, Scania, Embark Trucks, and others) developing their first prototypes to increase energy-efficiency [16]. Hence, the opportunity has risen to employ motion planning to optimise articulated vehicles’ velocity, driving path, and driving conditions for tyre wear minimisation.

There are some limited works on tyre management, focusing on human-driven high-performance race cars [17,18]. Tremlett et al. [17] investigated the optimal tyre management in a Formula 1 vehicle and its trade-off with lap time, proving that relatively small changes in control strategy can lead to significant reductions in the associated wear metrics. West et al. [18] further improved the tyre wear model [17] by adding tyre-carcass and tyre surface temperature as states and quantifying their effect on the total tyre wear. Both papers considered high-fidelity tyre wear models due to their focus on highly nonlinear tyre dynamics (i.e., high accelerations, increased tyre surface temperatures, high tyre frictional power, and others) and vehicle performance, which is not the case in passenger or heavy articulated vehicles except during evasive manoeuvres near the limit of handling. Meanwhile, none of these papers investigated the impact of driving style, vehicle capabilities, and lateral manoeuvrability on the trade-off between tyre wear and journey time. To the authors’ knowledge, there is no work considering the motion planning of automated articulated vehicles to decrease tyre wear, except some preliminary work by the authors [19].

The control of articulated and heavy vehicles to enhance their safety has been an issue in research. As a result, there are many works in the literature focusing on robust advanced driver assistance systems (ADASs) for articulated vehicles. For example, nonlinear lateral control algorithms, designed based on the multivariable backstepping technique for independently controlling the braking forces and for coordinated steering, can improve the lateral control of commercial heavy vehicles [20]. Similarly, the lateral stability and the tyre wear of an eight-axle vehicle were improved by controlling the direct yaw moment generated by the sixteen independent driving wheels [21]. In the same direction, active steering controllers for articulated vehicles have been developed to improve the path following of the rear part of the vehicle while driving on any path and at any velocity [15]. Other works [22] aimed at the minimisation of tyre wear through the optimisation of the command steering mechanism of an articulated vehicle using a genetic algorithm. Nevertheless, most of the existing works do not focus directly on minimising tyre wear, but the results illustrate a significant decrease in the lateral forces which the authors translate to less tyre wear due to limited scrubbing.

Apart from many works investigating the improvement of ADASs, researchers only recently started focusing more on fully automated heavy and articulated vehicles after the significant progress in automated passenger vehicles. A novel cascade path planning algorithm for automated truck–trailer parking was developed [23], while an adaptive path-following control for autonomous semi-trailer docking has been proposed in the literature [24]. Extending the work from the ADAS field, a robust path-following control for automated articulated heavy vehicles was prompted [25], which does not depend on the offline adjustment of tuning parameters and improved robustness, lateral stability, driving smoothness, and safety. Similarly, the optimal path planning for an automated articulated vehicle with two trailers was investigated [26] by combining the artificial potential field (APF) method and optimal control theory. At the same time, a motion planning system for heavy accelerating articulated vehicles was developed and tested in real road traffic scenarios [27]. More work is required that directly aims to minimise tyre wear through motion planning [19], such as the existing efforts to optimise the vehicle suspensions [12] or develop active control algorithms in this direction [28].

This paper focuses on formulating an optimal control problem (OCP) for optimising the trajectory of an articulated vehicle to minimise tyre wear and explore the fundamentals of trajectory planning for minimising tyre wear. More specifically, this paper investigates the following:How could a motion planning algorithm be designed for considering tyre wear minimisation in automated articulated vehicles?What is the trade-off between the journey time and the tyre mass loss due wear?To what extent can the vehicle’s powertrain and lateral manoeuvrability capabilities affect tyre wear?

To achieve this, an optimal control control problem is formulated. The optimum velocity profile is sought for a predefined road path from a specific starting point to a final one to minimise tyre wear at fixed time cases. Specific boundaries and constraints are applied to the problem to ensure the vehicle’s stability and the feasibility of the solution. The OCP is solved using GPOPS II, a MATLAB-based software for solving multiple-phase optimal control problems using hp-Adaptive Gaussian Quadrature Collocation Methods and Sparse Nonlinear Programming. More specifically, the OCP is solved using the direct approach from GPOPs [29] in combination with the NLP solver [30]. Through GPOPS II, a set of alternative optimal solutions are derived. Then, the OCP is modified to investigate how different driving styles, vehicle capabilities, and lateral manoeuvrabilities affect the set of alternatives. The results illustrate the trade-off between tyre mass loss due to wear and journey time, while they pave the way for AV researchers and manufacturers to consider tyre wear in their control modules and eventually come closer to the zero-emission goal.

This paper is organised as follows: firstly, the models (articulated vehicle and wear model) are described; secondly, the formulation of the optimal control problem is presented, including objectives, constraints, and boundaries; then, the results are described for the different scenarios studied; and, finally, conclusions are extracted.

## 2. Simulation Models

### 2.1. Articulated Vehicle Model

To model an articulated vehicle, several aspects needs to be considered. To minimise the computational time of the optimal control problem, the articulated vehicle model should be simple. Moreover, since the focus is on tyre wear, it is necessary to include slip angles and forces in the model. Therefore, a dynamic model is required rather than a simple kinematic model. Based on this, the chosen articulated vehicle model (AVM) is a linear single-track three-degrees-of-freedom bicycle model, which includes the lateral and yaw dynamics of the tractor and the trailer and their coupling dynamics [20,31] (Figure 1a).

The AVM parameters were extracted from IPG/TruckMaker for the model to represent a Volvo FH-500 2012 (Figure 1b). The equations are presented below (Equations (Equation 1)–(Equation 7)) as extracted from the literature [20]. Equation (Equation 1) represents the sum of the lateral forces applied to the full body (i.e., tractor and trailer).
(1)$$\left(m_{1}+m_{2}\right){\dot{v}}_{y}-m_{2}\left(d_{1}+d_{3}\right)\ddot{\psi }-m_{2}d_{3}\ddot{\theta }+\left(m_{1}+m_{2}\right)v_{x}\dot{\psi }=F_{yf}+F_{yr}+F_{yt}$$
where $\dot{\psi }$ is the tractor yaw rate, and $\ddot{\psi }$ is the tractor yaw acceleration; ${\dot{v}}_{y}$ is the derivative of the lateral velocity; $\ddot{\theta }$ is the hitch rotational acceleration; $v_{x}$ is the longitudinal velocity; $m_{1}$(=6800 kg) is the tractor mass and $m_{2}$(=10,350 kg) is the unloaded trailer mass; $d_{1}$(=1.57 m) is the distance from the hitch point to the truck center of gravity (COG); $d_{3}$(=5.00 m) is the distance from the hitch point to the trailer COG; and $F_{yi}$ is the lateral tyre force of the *i*th axle, i.e., the front tractor (*f*), rear tractor (*r*), and trailer (*t*) axle. Equations (Equation 2) and (Equation 3) correspond to the sum of the moments applied to the whole body as calculated around the COG of the tractor and the sum of the moments applied to the trailer around the axis of the hitch angle.
(2)$$\begin{array}{c} -m_{2}\left(d_{1}+d_{3}\right){\dot{v}}_{y}+\left(I_{z1}+I_{z2}+m_{2}{\left(d_{1}+d_{3}\right)}^{2}\right)\ddot{\psi }+\left(I_{z2}+m_{2}d_{3}^{2}+m_{2}d_{1}d_{3}\right)\ddot{\theta }\\ -m_{2}\left(d_{1}+d_{3}\right)v_{x}\dot{\psi }=F_{yf}l_{1}-F_{yr}l_{2}-F_{yt}\left(d_{1}+l_{3}\right) \end{array}$$
(3)$$-m_{2}d_{3}{\dot{v}}_{y}+\left(I_{z2}+m_{2}d_{3}^{2}+m_{2}d_{1}d_{3}\right)\ddot{\psi }+\left(I_{z2}+m_{2}d_{3}^{2}\right)\ddot{\theta }-m_{2}d_{3}v_{x}\dot{\psi }=-F_{yt}l_{3}$$
where $I_{z1}$(=13,000 kg/$m^{3}$) and $I_{z2}$(=48,280 kg/$m^{3}$) are the yaw inertia of the tractor and the trailer, respectively; $l_{1}$(=1.05 m) and $l_{2}$(=2.5 m) are the distances between the tractor COG and the front and rear axle, respectively; and $l_{3}$(=7.75 m) is the distance between the hitch point and the trailer axle. The lateral forces are considered linear and are calculated as the product of the cornering stiffness ($C_{a}$) and the sideslip angle (α) of the corresponding axle as illustrated in Equation (Equation 4):(4)$$F_{yf}=2C_{af}\alpha_{f},\;\;\;F_{yr}=2C_{ar}\alpha_{r},\;\;\;F_{yt}=2C_{at}\alpha_{t}$$
where $C_{af}$(=252,000 N/rad), $C_{ar}$(=236,000 N/rad), and $C_{at}$(=263,500 N/rad) are the cornering stiffnesses of the front tractor, rear tractor, and trailer axle, respectively; $\alpha_{f}$, $\alpha_{r}$, and $\alpha_{t}$ are the sideslip angles of the corresponding axles. Assuming small angles, the sideslip angles ($\alpha_{f}$, $\alpha_{r}$, and $\alpha_{t}$) can be reformulated [20,31]:(5)$$\alpha_{f}=\delta_{1}-\frac{v_{y}+l_{1}\dot{\psi }}{v_{x}},\;\;\alpha_{r}=-\frac{v_{y}+l_{2}\dot{\psi }}{v_{x}},\;\;\alpha_{t}=\theta -\frac{v_{y}-d_{1}\dot{\psi }-l_{3}\left(\dot{\psi }+\dot{\theta }\right)}{v_{x}}$$

Combining Equations (Equation 1)–(Equation 5), the final equations of motion can be derived in matrix form as follows:(6)$$M\ddot{x}+C\left(x,\dot{x}\right)+D\dot{x}+Kx=F\delta$$
where x=[y,ψ,θ] is the state vector and δ is the input vector. The inertial (*M*), the centrifugal and Coriolis force (*C*), the damping (*D*), the potential (*K*), and the force (*F*) matrices can be found in the literature [20].
(7)$$\ddot{x}=M^{-1}F\delta -M^{-1}C\left(x,\dot{x}\right)-M^{-1}D\dot{x}+M^{-1}Kx$$

### 2.2. Curvilinear Coordinates

Given that the journey time will be included in the formulation of the optimal control problem, the equations of motion should be defined in the space domain. In this way, the behaviour of the articulated vehicle is simulated along the vehicle position by removing the equation’s dependence on time. Hence, the time could be included in the formulation of the problem either as a constraint or as an objective. To that end, the transformation of the model in the curvilinear coordinates [19] is conducted as follows:(8)$${\dot{s}}_{s}=\frac{v_{x}cos\left(\alpha_{s}+\beta \right)-v_{y}sin\left(\alpha_{s}+\beta \right)}{1-s_{n}\kappa }$$
(9)$${\dot{s}}_{n}=v_{x}sin\left(\alpha_{s}+\beta \right)+v_{y}cos\left(\alpha_{s}+\beta \right)$$
(10)$${\dot{\alpha }}_{s}=\dot{\psi }-\kappa \frac{v_{x}cos\left(\alpha_{s}+\beta \right)-v_{y}sin\left(\alpha_{s}+\beta \right)}{1-s_{n}\kappa }$$
(11)$$\beta =\frac{v_{y}}{v_{x}}$$
where $s_{s}$ is the vehicle position; $\alpha_{s}$ is the relative orientation compared to the road orientation; $s_{n}$ is the lateral off tracking from the centerline; β is the vehicle sideslip angle; and κ is the local curvature of the road. The transformation in the curvilinear coordinates is implemented using the chain rule:(12)$$\begin{array}{c} \frac{d\xi }{ds_{s}}=\xi^{\prime }=\frac{d\xi }{dt}\frac{dt}{ds_{s}}\\ \frac{dt}{ds_{s}}=\frac{1}{{\dot{s}}_{s}}\Rightarrow \xi^{\prime }=\dot{\xi }{\frac{1}{\dot{s}}}_{s} \end{array}$$
where ξ is any state; ${\dot{s}}_{s}$ is defined according to Equation (Equation 8). The transformation is applied in every control input and state. As the equations are redefined in the space domain, time can now be defined as a state and is calculated as follows:(13)$$\frac{dt}{ds}=\frac{1}{{\dot{s}}_{s}}=t^{\prime }$$

Finally, the global coordinates of the road centerline are extracted from data that describe the global coordinates ($x^{\prime }$,$y^{\prime }$, and κ) of the road centerline.

### 2.3. Wear Model

The sideslip (α) and the longitudinal slip (*s*) are directly related to the tyre mass loss due to wear. Sideslip (α, Equation (Equation 5)) is the angle between the pointing direction of the wheel and its actual direction of travel, and generates a lateral force ($F_{y,j}$) that deforms the rubber wearing down the tyre. The longitudinal slip (Equation (Equation 14)) is the difference between the rotational velocity of a point on the tyre ($\omega \;R_{\omega }$) and the vehicle velocity ($v_{x}$).
(14)$$s=\frac{\omega R_{\omega }-v_{x}}{v_{x}}$$

In cases where the wheel rotates with a velocity greater or smaller than the free-rolling velocity ($\omega R_{\omega }\neq v_{x}$), the tread layers illustrate different velocities. As a results, shear stress is generated, and eventually more tyre deformations and wear occur.

The calculation of tyre wear can be conducted through the tyre dissipated energy, which can be evaluated through the frictional power in the longitudinal ($P_{x}$) and the lateral ($P_{y}$) direction according to Equation (Equation 15):(15)$$\begin{array}{cc} P_{x,j}=F_{x}v_{x}s & \\ P_{y,j}=F_{y}v_{x}\alpha \end{array}$$
where j=f,r, and *t* represent the tyre position. The frictional power per unit area (*W*) can be evaluated in both directions ($W_{x}$ and $W_{y}$) according to Equation (Equation 16) by dividing the frictional power with the contact area between the tyre and the road ($A_{j,patch}$):(16)$$\begin{array}{cc} W_{x,j}=P_{x,j}/A_{j,patch} & \\ W_{y,j}=P_{y,j}/A_{j,patch} \end{array}$$
(17)$$A_{j,patch}=2nD_{patch}\sqrt{R_{0}^{2}-R_{j,Lo}^{2}}$$
where $D_{patch}$ is the tyre width; *n* is the tread pattern proportion; $R_{0}$ is the tyre radius; and $R_{j,Lo}$ is its loaded radius. The nominal radius is extracted from IPG/TruckMaker 13, while the $R_{j,Lo}$ is calculated according to
(18)$$R_{j,Lo}=R_{0}-F_{z}/K_{s}$$
where $K_{s}$ = 1.27 × $10^{6}$ N/m is the tyre vertical stiffness according to IPG/TruckMaker 13. Based on the frictional power per unit area, the calculation of the tyre mass loss due to wear (ΔmL) per unit area is possible using the equation both in the longitudinal and lateral directions:(19)$$\begin{array}{cc} \Delta mL_{x,j}=k_{1}\;W_{x,j}^{k_{2}} & \\ \Delta mL_{y,j}=k_{1}\;W_{y,j}^{k_{2}} \end{array}$$
where $k_{1}$(=1.0 × $10^{-10}$) and $k_{2}$(=1.74) are the constants relating tyre mass loss due to wear and its frictional power [32].

### 2.4. Limitations of the Selected Models

The vehicle model as well as the tyre model were chosen based on a number of considerations. The road surface was considered smooth; therefore, no vibrations were induced to the vehicle body. Thus, the chosen vehicle model does not consider vertical dynamics, neglecting the influence of the vertical tyre–road interaction and the role of the suspension systems. At the same time, since in normal driving conditions no significant thermal phenomena are expected, the abrasive mechanism, used to model tyre wear, is a simplification and does not consider degradation due to thermal or chemical fatigue phenomena. An additional assumption considered is that the tyre wear model neglects interaction effects between lateral and longitudinal dynamics. This relies on the fact that the driving conditions studied in this work are normal and not at the limit of handling. The chosen type of articulated heavy vehicle has one driving axle, i.e., the rear tractor axle, which means the vehicle model only considers longitudinal slip in this axle. Meanwhile, no braking system is employed in the trailer to generate more longitudinal slip. Hence, the combined slip events are limited and the configuration of the optimal control problem will limit them further to secure normal driving conditions.

## 3. Optimal Control Problem (OCP)

When a route and manoeuvre are selected from route planning, the AV needs to investigate the specifics on how to (optimally) perform the manoeuvre. So, trajectory optimisation is employed to find the optimal trajectory that minimises a certain performance aspect or state with respect to path constraints. In this work, the trajectory optimisation is performed through the formulation of an optimal control problem (OCP). In the OCP, the vehicle starts from an initial state and ends at a final state, while between these states the path that the AV is to follow is predefined. While on this path, the AV is constrained to the road dimensions (i.e., width and length), while the control inputs are optimised for the vehicle to be controlled based on the objectives and various constrains. The OCP aims to optimally control the articulated vehicle to minimise tyre wear without compromising journey time.

### 3.1. System Dynamics, Control Inputs, and States

The dynamic equations of the vehicle (Equations (Equation 7)–(Equation 10)) are formulated as follows:(20)$$\dot{x}\;\;=\;f\left(x,u\right)$$
where *x* is the state, and *u* is the control input. The equations are converted to the distance domain according to Equations (Equation 12) and (Equation 13):(21)$$x^{\prime }=t^{\prime }f\left(x,u\right)$$

The control inputs (*u*) selected are the longitudinal tyre force rate (${\dot{F}}_{x,j}$) and the steering rate ($\dot{\delta }$):(22)$$u=[{\dot{F}}_{x,j},\dot{\delta }]$$
where $F_{x,j}$ is the longitudinal tyre force at the *j*th axle; δ is the steering wheel angle. Both are included as state variables. Regarding the longitudinal forces, the front tractor ($F_{xf}$) and trailer ($F_{xt}$) axle forces are zero (i.e., no traction is provided), while the rear tractor axle is only providing traction ($F_{xr}\neq$0).

By including the control inputs as state variables, the tractor longitudinal acceleration ($a_{x}$) can be calculated as follows:(23)$$a_{x}=\frac{\Sigma F_{x}}{m_{1}+m_{2}}$$
where the $\Sigma F_{x}$ is calculated based on the sum of the tyre longitudinal forces, i.e., only $F_{xr}$. Then, according to $a_{x}$, the longitudinal velocity ($v_{x}$) is defined. This assumption (Equation (Equation 23)) is considered to include simplified longitudinal dynamics in the AVM rather than a constant velocity. Finally, the state vector consists of sixteen variables as illustrated in Equation (Equation 24):(24)$$x=\left[\psi ,\theta ,v_{y},\dot{\psi },\dot{\theta },v_{x},\delta ,s_{s},s_{n},\alpha_{s},x,y,t,F_{x}\right]$$

### 3.2. Objectives

The aim of the OCP is to optimally plan the trajectory of an articulated vehicle in the predefined path in order to minimise tyre wear without compromising journey time. Thus, a multi-objective optimisation problem is formed for the trade-off of the two objectives to be explored. To calculate the total tyre mass loss due to wear, i.e., the first objective, the product of Equation (Equation 19) with the tyre width in contact with the road is evaluated. Then, the product is integrated over the complete path as follows:(25)$$\begin{array}{c} m_{y,j}=\int_{s_{0}}^{s_{f}}\left(2\cdot \eta \cdot D_{patch}\cdot \Delta mL_{y,j}\right)dS\\ m_{x,j}=\int_{s_{0}}^{s_{f}}\left(2\cdot \eta \cdot D_{patch}\cdot \Delta mL_{x,j}\right)dS \end{array}$$
where $s_{0}$ and $s_{f}$ are the initial and final points in the road. The tyre mass loss due to wear as occurred in each vehicle axle is summed, producing the cost function for the OCP.
(26)$$\begin{array}{c} J_{1y}=\sum_{j=1}^{3}m_{y,j}\\ J_{1x}=\sum_{j=1}^{3}m_{x,j}\\ J_{1}=J_{1y}+J_{1x} \end{array}$$

As mentioned before, the vehicle only has traction on the rear tractor wheels and the others are all free-rolling. Thus, they present zero longitudinal slip, i.e., the longitudinal frictional power is zero for the free-rolling tyres, leading to zero longitudinal mass loss ($mL_{x,f}=mL_{x,t}=0$).

In addition to tyre wear, journey time is also included as an objective in order to not be compromised while tyre wear is minimised. The cost function can be derived as an integral over Equation (Equation 13):(27)$$J_{2}=\int_{s_{0}}^{s_{f}}\frac{1}{{\dot{s}}_{s}}dS$$

According to the previous section, the optimisation problem is a multi-objective one and, to transform the problem in a single objective problem, the epsilon constraint method (ECM) is employed. The ECM selects one objective as the main one, and expresses the others as inequality constraints. The benefits gained with the employment of the ECM compared to the consideration of weights is that the ECM delivers optimal results regardless of if the set is convex or non-convex. At the same time, no normalisation or a priori knowledge of the weighting factors is required for the objective functions. In this work, the ECM is used with $J_{1}$ as the objective to be minimised, and $J_{2}$ as the one to be constrained, i.e., the OCP seeks for the optimal trajectory to minimise $J_{1}$ at a fixed time solution ($J_{2}=T_{fixed}$). In order to produce multiple optimal alternatives, $J_{2}$ is varied. By obtaining the multiple alternatives, a pareto front illustrating the conflict between tyre mass loss due to wear and journey time is produced. To sum up, the OPC is formulated as follows:(28)$$OCP=\left\{\begin{array}{c} min\;J\\ \;\\ s.t.:|J_{2}|<T_{fixed}\\ x^{\prime }\left(s_{s}\right)=f\left(s_{s},x\left(s_{s}\right),u\left(s_{s}\right)\right)\\ x\left(s_{s_{0}}\right)=x_{initial},\;x\left(s_{s_{f}}\right)=x_{final}\\ \;\\ x\left(s_{s}\right)\leq x_{max}\\ u\left(s_{s}\right)\leq u_{max}\\ x\left(s_{s}\right)\geq x_{min}\\ u\left(s_{s}\right)\geq u_{min}\\ h(s_{s},x\left(s_{s}\right),u\left(s_{s}\right))\leq 0 \end{array}\right\}.$$
where h($s_{s}$, x($s_{s}$), and u($s_{s}$)) are the path constraints that will be presented later in detail; $T_{fixed}$ varies depending on the simulation. More specifically, ten fixed time solutions are considered to seek their optimal trajectory:(29)$$T_{fixed}=\left[300\;\;310\;\;320\;\;330\;\;340\;\;350\;\;360\;\;370\;\;380\;\;390\right]\;\;s$$
where, from these alternatives, the first case ($T_{fixed}=300\;s$) is the minimum time solution that corresponds to the selected road path, which is presented later. The velocity ($v_{x}$) was bounded to be from 25 km/h to 90 km/h, while the initial velocity was set at 30 km/h.

### 3.3. Constraints

#### 3.3.1. Lateral and Longitudinal Accelerations

The limit for the lateral acceleration is set at 0.4 g to secure that no rollover occurs in curves. This limit is selected according to ECE-R111 [33]. Therefore, the following constraint is considered:(30)$$|a_{y}|\leq 0.40\;g$$

To further ensure that the lateral acceleration is at acceptable levels, the steady-state lateral acceleration for the trailer and the truck are also constrained with Equations (Equation 31) and (Equation 32). This secures that the combination of the yaw rate and the velocity do not result in high lateral acceleration peaks.
(31)$$\begin{array}{c} |\dot{\psi }v_{x}|\;\;\;\;\;\;\;\;\leq \;0.40\;g \end{array}$$
(32)$$\begin{array}{c} |\left(\dot{\theta }-\dot{\psi }\right)v_{x}|\leq \;0.40\;g \end{array}$$

The upper limit of the longitudinal acceleration ($a_{x}$) is mainly dependent on the vehicle’s powertrain capabilities. The lower limit is related to the vehicle’s braking capabilities, where load and maximum brake force are considered. The absolute value of the lower limit is much higher than the upper limit of the longitudinal acceleration. Thus, the following constraints are considered for the longitudinal acceleration:(33)$$-0.25\;g\leq a_{x}\leq ax_{max}$$
where $ax_{max}$ in the default case is 0.03 g. The low $ax_{max}$ values secure the minimisation of the longitudinal slip in the driven axle, avoiding combined slip events that the simplified model used in this work cannot capture.

#### 3.3.2. Steering and Hitch Angle

To ensure that the vehicle does not steer too much, the steering angle is constrained. The vehicle model and the tyres are linear; hence, the steering should be bounded in the linear region. Large steering angles result in nonlinearity and eventually instability of the vehicle, and potentially in rollover and snaking. Therefore, according to the literature [34], the steering angle is constrained based on Equation (Equation 34). In addition, the steering angle should not change abruptly to ensure that the trajectory is smooth. For this, the steering rate is also constrained (Equation (Equation 35)):(34)$$\begin{array}{c} |\delta |\leq \pi /9\;\;\mathrm{rad}\;\; \end{array}$$(35)$$\begin{array}{c} |\dot{\delta }|\leq 0.30\;\mathrm{rad}/s \end{array}$$

The hitch angle is another important aspect to limit, as high values increase the risk of jack knifing. Therefore, the hitch angle is limited to stay within a small interval (Equation (Equation 36)). Moreover, the hitch rate is constrained to secure a smooth trailer movement (Equation (Equation 37)).
(36)$$\begin{array}{c} |\theta |\leq \;\pi /6\;\mathrm{rad}\; \end{array}$$
(37)$$\begin{array}{c} |\dot{\theta }|\leq 0.30\;\mathrm{rad}/s \end{array}$$

The authors used these values based on offline simulations with the commercial software IPG/TruckMaker and the literature [35,36].

#### 3.3.3. Prevention of Skidding and Sliding

The slip angles should be bounded to secure the linearity of the tyre dynamics, so Equation (Equation 38) is employed:(38)$$|\alpha_{i}|\leq \pi /32\;\mathrm{rad}$$
where i=f,r, and *t*, depending on the axle. Additionally, for stability, large values of lateral velocity should be avoided to avoid skidding, sliding, and other events. For this, the vehicle sideslip angle (β) is constrained (Equation (Equation 39)) to bound the lateral velocity in relation to the longitudinal one. Typical values of vehicle sideslip angles are around π/90 rad, and they provide dynamically stable driving.
(39)|β|≤π/90rad

The authors used these values based on offline simulations with the commercial software IPG/TruckMaker 13 and the literature [35,36].

#### 3.3.4. Road Path and Borders

The vehicle has to travel within the road’s limits, so the road boundaries must be defined as constraints. First of all, the vehicle should start from the requested point ($s_{start}$) and reach the final point ($s_{final}$) of the road (Equation (Equation 40)). Then, during this part, the vehicle should be within the left and right road borders (Equation (41)). If the road width (RW) is considered to be around 4 m and the vehicle width (VW) to be 2 m, then the lateral off tracking ($s_{n}$) of the vehicle’s center of gravity (CoG) should be bounded according to Equation (41).
(40)$$\begin{array}{c} s_{start}\leq s_{s}\leq s_{final} \end{array}$$
(41)$$\begin{array}{c} \frac{-RW+VW}{2}\leq s_{n}\leq \frac{RW-VW}{2} \end{array}$$

## 4. Results and Discussion

The OCP, as defined in the previous sections, is solved first for multiple fixed time cases ($T_{fixed}$) to investigate the conflict between tyre mass loss due to wear and journey time for different driving styles. Then, the OCP is re-configured to investigate how the trade-off was affected for different vehicle capabilities and lateral maneuverability.

### 4.1. Scenarios for Analysis

The scenarios considered are as follows:Driving style: The optimal solution is sought for multiple fixed time cases (Equation (Equation 29)), allowing us to investigate and quantify the trade-off between the tyre mass loss due to wear and journey time. This setup is the default one with the configuration presented in Section 3. This analysis focuses on defining the trade-off between tyre wear and journey time for future automated articulated vehicles. In this way, when an articulated vehicle is late or on time for delivery to a point, the company could shift to another driving style for the next point to decrease the tyre wear or carefully request a faster journey time considering the environmental impact.Vehicle capabilities: The optimal solutions obtained for various fixed journey time cases in the scenario are re-run after varying the maximum vehicle longitudinal acceleration ($ax_{max}$, Equation (Equation 34)). The values considered are 0.03 g, 0.05 g, and 0.07 g, and are selected according to the acceleration capability performance levels proposed by the Australian Transport Council ([37]). The configuration with 0.03 g is the default one used in the driving style scenario. The current scenario focuses on investigating if an articulated vehicle with higher powertrain capabilities, i.e., with the ability to provide greater longitudinal accelerations, could affect the tyre mass loss due to wear.Vehicle lateral maneuverability: The optimal solutions obtained for various fixed journey time cases in the first scenario are re-run after varying the road width (RW), which affects the lateral off tracking from the centerline (Equation (41)). The values considered for the road width are 4.0, 3.5, and 3.0 m, and these limit the lateral off tracking from the centerline ($s_{n}$) to 1.00, 0.75, and 0.5 m, respectively. The configuration with a 4.0 m road width is the default one used in the driving style scenario. This scenario focuses on studying how the tyre mass loss due to wear will be affected due to different national road designs or an unexpected event, i.e., if another vehicle enters the lane.

In the last two scenarios, the $T_{fixed}$ cases are studied from 310 to 390 s, as the various modifications affect the minimum time, which in some cases was less or more than 300 s. Regarding the road path used for the trajectory planning, it is around 6.0 km long and consists of countryside road characteristics as shown in Figure 2. The curvature of the road ranges from −0.025 to 0.025 ($m^{-1}$), illustrating a radius down to 40 m during cornering. No road inclination and roughness are considered, while obstacles and disturbances are neglected so we could study a collision-free path. Later in the analysis, the focus is on the first one or two kilometers as illustrated in Figure 2b, to present the optimal states, control inputs, and pat with more clarity.

### 4.2. Optimal Values of the Objectives

As mentioned before, for the first scenario, the OCP is solved for multiple fixed time cases ($J_{2}=T_{fixed}$, Equation (Equation 29)). The multiple solutions obtained from GPOPs’s solver allows for the generation of the pareto front of the problem (Figure 3) and the quantification of the conflicting relation between tyre mass loss due to wear and journey time. More specifically, Figure 3 illustrates the trade-off between the objectives, i.e., total tyre mass loss due to wear ($J_{1}$, Equation (Equation 26)) and journey time ($J_{2}$, Equation (Equation 27)), of the optimal solutions obtained. Moreover, the optimal solutions are simulated in IPG/TruckMaker 13 in order to evaluate the objectives with a higher fidelity articulated vehicle model and then compare them with the ones obtained from the trajectory planning. These two sets of alternatives are further investigated in Table 1, where the decrease in the tyre mass loss due to wear ($J_{1}$) per increase in the journey time ($J_{2}$) is calculated for both.

Based on the results, there is a clear conflict between the tyre mass loss due to wear and the journey time as depicted in the optimal solutions. According to Figure 3, the increase in $J_{2}$ leads to a decrease in $J_{1}$. More specifically, based on Table 1, the increase in $J_{2}$ in the OCP from one $T_{fixed}$ case to the next (∼3% for each) leads to an average decrease in $J_{1}$ of around 28.8%. Similar decreases (∼35.3%) are illustrated when the solutions are simulated in IPG/TruckMaker 13. This relation is important because it illustrates how the demand for a relatively small increase in the journey time leads to such a significant decrease in the tyre mass loss due to wear. A greater decrease in $J_{1}$ is identified when $J_{2}$ is increased from 300 to 310 s, i.e., from the minimum time solution to a more flexible one (i.e., Solution 1). This is because the minimum time solution is pushing the vehicle to its limits and even a small increase in journey time significantly smooths the driving style, which hence decreases the tyre mass loss due to wear to a great extent.

Regarding the comparison of the objective values of the optimal solutions as obtained from the solver and after their simulation in the software (Figure 3), the results have significant differences in the magnitude, 26.6% on average. The largest difference is identified in the minimum time solution and the smallest one in Solution 3. These differences are due to the AVM used in the formulation of the OCP, which is a linear vehicle model with a linear tyre model built mainly for lateral control studies. On the one hand, the AVM is not considering tyre nonlinearities around the minimum time solution where they might emerge, and this could lead to an underestimation of the tyre wear, which depends on the longitudinal slip (slip ratio) and lateral slip angle. Moreover, neglecting the vertical dynamics may have an impact on this underestimation as well. Therefore, for these reasons the differences between the objectives around Solution 1 are higher. On the other hand, close to Solution 3, the driving behaviour has smoothened significantly and the obtained solutions are similar to the simulated ones. However, regardless of the differences, the trend in the pareto front is captured successfully, while decreases in $J_{1}$ with the increase in $J_{2}$ from one $T_{fixed}$ case to the next are similar according to Table 1. This is also depicted in the average value of the decrease in $J_{1}$, which is 28.8% and 35.3% for the optimal solutions as obtained from the solver and after their simulation in IPG/TruckMaker 13, respectively.

### 4.3. Optimal States and Control Inputs

From the optimal solutions obtained, three are selected for further analysis regarding their optimal states and control inputs. Regarding the selected solutions for further analysis, Solution 1 is the closest to the minimum journey time solution of this path, Solution 2 is a compromise of the two objectives, and Solution 3 is mostly for tyre wear minimisation. For these three solutions, their optimal velocity profile (Figure 4) and path characteristics (Figure 5) are analysed. As far as the path characteristics are concerned, Figure 5a illustrates the tractor’s center of gravity (CoG) lateral off tracking ($s_{n}$) from the centerline, which is bounded according to Equation (41). Furthermore, Figure 5b presents the trajectory of the tractor’s CoG. The lateral vehicle displacement (Y-axis) is plotted versus the distance travelled instead of the longitudinal displacement. Regarding the control inputs, the algorithm seeks the optimal rate of the steering angle ($\dot{\delta }$) and the longitudinal force ($\dot{F_{x}}$). However, in Figure 6, the corresponding optimal steering angle (δ) and longitudinal force ($F_{x}$) are presented to be able to interpret them physically. For the analysis in this section, the focus is on the first kilometer of the road path as illustrated in Figure 2b.

As far as the optimal solutions are concerned, according to Figure 4, their velocity profiles illustrate a similar trend, i.e., acceleration and deceleration at the same points of the path, but with different magnitudes. Solution 1 is the one with the higher magnitude velocity, where the max velocity is reached for the first time around 2.5 km and then again at 3 km for around 1.0 km. This occurs because Solution 1 aims to arrive at the final destination as close to the minimum journey time as possible. Then, Solution 3 is the one with the lowest velocity magnitudes as $T_{fixed}$ in this solution is the maximum among the alternatives. The compromise of these two solutions, i.e., Solutions 1 and 3, is Solution 2, which illustrates parts where velocity is at the upper limit (90 km/h), as in Solution 1, and parts with lower velocity, as in Solution 3.

Regarding the path characteristics, from both figures in Figure 5, Solution 1 is cutting corners as much possible to gain time and satisfy the objective of a shorter journey time. More specifically, in Figure 5a, $s_{n}$ is positive for Solutions 1 and 2 from 0.20 to 0.28 km, whereas for Solution 3 it is negative. This implies that Solutions 1 and 2 are cutting off this corner, while Solution 3 is widening it. Furthermore, Solution 1 leads the vehicle to drive at the left road border for this corner pushing $s_{n}$ at the maximum value for this scenario, i.e., 1 m, while Solution 2 also drives at the maximum allowed $s_{n}$ but for a shorter distance. The difference is that the transition to the road border is smoother in Solution 2 to also decrease the tyre mass loss due to wear. These remarks are also depicted in Figure 5b, where the corner cutting-off is also visible from the trajectory.

Regarding the control inputs (Figure 6), the results are in accordance with the optimal path (Figure 5) and the velocity profile (Figure 4). More specifically, the optimal steering angle in the three solutions is around the same levels of magnitude as they are optimised for the same path. However, Solution 1 illustrates more abrupt changes in the corners and this is depicted in the larger peaks (like the ones around 0.2, 0.4, 0.55, 0.8, and 0.9 km) compared to the other solutions. Furthermore, different driving behaviours and the cutting of corners can be identified. For example, in the beginning of the area of focus (0.15–0.30 km, Figure 5b), the steering angle of Solutions 1 and 2 (Figure 6a,b) is illustrating contradicting behaviour with Solution 3. This is why Solutions 1 and 2 are cutting off the corner to the left of the centerline, while Solution 3 is widening it to the right of the centerline (Figure 5b).

As far as the longitudinal forces are concerned (Figure 6c), in the first part of the first kilometer, where the emphasis is on the current analysis, all the solutions are applying the maximum allowable longitudinal force as they seek to increase their initial velocity to the optimal levels. Solution 1 applies these levels of longitudinal forces for a longer distance as the optimal levels of the velocity are higher than the others to reach the final destination in close to the minimum time. Additionally, Figure 6c validates the remark for Figure 4 regarding more abrupt acceleration and deceleration events. More specifically, more abrupt increases or decreases in the longitudinal forces, i.e., accelerating or braking events, respectively, take place for Solution 1. For example, for the braking event between 0.6 and 0.7 km, Solution 1 requires around 0.05 km to reach the maximum braking force, whereas Solution 3 initiates the braking before 0.4 km to peak the braking force between 0.6 and 0.7 km. This occurs similarly in Solution 2, but the braking is initiated from 0.5 km. After this event, the main difference between these solutions is in the peak magnitude as there are few consecutive directional changes.

### 4.4. Vehicle Capabilities and Lateral Maneuverability

As mentioned before, two additional scenarios were investigated to study the impact of different vehicle capabilities and lateral maneuverability on the trade-off between the two objectives. The configuration of the OCP used in the driving style scenario was modified for the two additional scenarios. For each scenario, two modifications, namely two different maximum longitudinal accelerations and road widths, were simulated, leading to the provision of two pareto fronts for each scenario. The results are illustrated in Figure 7, where the pareto front of the first scenario, i.e., driving style, is compared with the pareto fronts obtained after modifying the OCP. Moreover, the impact that each configuration had on every optimal solution is evaluated for both scenarios in Table 2 and Table 3. Regarding the vehicle capabilities analysis, a further investigation was conducted on the vehicle states to justify the outcome. More specifically, the velocity, the longitudinal acceleration, and the lateral acceleration of the optimal solutions at $T_{fixed}=$ 320 s for the three scenarios ($a_{x,max}=$0.03 g, 0.05 g, and 0.07 g) were compared.

According to the results, when the vehicle capabilities are increased gradually, i.e., $ax_{max}$ is varied from 0.03 g to 0.05 g or 0.07 g, the tyre wear is decreased significantly. This implies that the employment of articulated vehicles with low powertrain capabilities will lead to greater tyre mass losses due to wear if the journey time requirements are maintained at the same levels as vehicles with greater powertrain capabilities. The reason is that the vehicle achieves the same velocity and journey time requirements without operating for long parts of the journey at the maximum acceleration limits. Furthermore, the jerk of the acceleration, i.e., their rate, is smoother. This remark can be validated by Figure 8. According to it, the vehicle velocity (Figure 8a) is higher when $ax_{max}$ is lower, as the vehicle needs to maintain a higher velocity for a longer time to reach the final destination at the same time as a vehicle with higher powertrain capabilities. This leads the optimal solution to maintain the vehicle longitudinal acceleration ($a_{x}$) at $ax_{max}$ levels for a longer time compared to the other scenarios (Figure 8b). At the same time, the lower longitudinal acceleration limits and the same journey time requirements lead to an increase in the lateral acceleration ($a_{y}$), as depicted in Figure 8c, which eventually leads to much more wear. This contradictory behaviour, i.e., increasing $a_{x}$ and decreasing $a_{y}$ as occurs around 1.7 km, illustrates that the optimal solution is avoiding combined slip effects as these are not maximised simultaneously.

According to Table 2, there is no need for an excessive increase in powertrain capabilities as after some point the decrease in tyre wear is saturated. More specifically, the average value of the decrease in $J_{1}$ from the main driving style scenario is 23.6% and 26.5% for each modification, respectively. Based on this, it is obvious that the increase in $ax_{max}$ from 0.03 g to 0.05 g decreases tyre wear more significantly, while the increase in $ax_{max}$ from 0.05 g to 0.07 g offers minor but not insignificant decreases. However, the modification of a vehicle to reach these powertrain capabilities might not be cost-efficient for that level of tyre wear decrease. Finally, the level of difference between the main driving style solution and the modified scenario solutions gradually decreases from the first to the last $T_{fixed}$ case.

In addition to the vehicle capabilities, the lateral maneuverability, i.e., the decrease in RW from 4.0 m to 3.5 or 3.0 m, is studied with regard to its impact on tyre wear. According to the results (Figure 7), the decrease in the road width leads to the increase in the tyre mass loss due to wear. The decrease in the road width increases the optimal path instantaneous road curvature (κ) at the average rate, as on the one hand the road centerline and its curvature is unchanged, but on the other hand the optimal path is forced to be closer to the centerline due to the limited lateral maneuverability. This decrease leads to the increase in the lateral tyre forces ($F_{y}$) through the increase in the lateral acceleration ($a_{y}=v_{x}^{2}\kappa$ at steady-state conditions), which, eventually, leads to more tyre wear. Based on Table 3, the variation in the lateral maneuverability has affected tyre wear levels less compared to the vehicle capabilities. This is because the average value of increase in $J_{1}$ from the main driving style scenario is 9.0% and 20.4% for each lateral maneuverability modification, respectively, whereas for the vehicle capabilities it is around 25% for both. The average values outline that the modifications have a similar impact on $J_{1}$ for the various $T_{fixed}$ cases, but the decrease in RW from 4.0 to 3.5 m increases tyre wear less than the decrease in RW from 3.5 to 3.0 m. Finally, the level of difference between the main driving style solution and the modified scenario solutions is similar for all the $T_{fixed}$ cases, and only the first two (i.e., 310 and 320 s), which are close to the minimum time, are affected more.

## 5. Conclusions

To sum up, this work investigates the fundamentals of motion planning for tyre wear minimisation by assessing how different driving styles, vehicle capabilities, and lateral maneuverability affect the optimal solutions.

In summary, the following conclusions are extracted to answer the corresponding research questions raised in the Introduction:This paper presented a configuration of an optimal control problem which optimises the trajectory of an articulated vehicle for tyre wear minimisation. The significant minimisation of the tyre wear achieved through the optimal trajectory was validated through simulations with a commercial software.The increase in journey time, even of an insignificant amount (∼3%), leads to an average decrease in tyre mass loss due to wear of around 29%. This relation is important because it illustrates how a relatively small increase in the journey time leads to such a significant decrease in the tyre mass loss due to wear.The employment of articulated vehicles with low powertrain capabilities leads to greater mass losses due to tyre wear if the journey time requirements are maintained at the same levels as vehicles with greater powertrain capabilities. At the same time, there is no need for excessive increases in powertrain capabilities as after some point the decrease in tyre wear is saturated. The decrease in the vehicle lateral maneuverability leads to the increase in the tyre mass loss due to wear as expected. More specifically, if the road width is decreased by 0.5 m, the tyre mass loss due to wear can be increased by around 10%.

The above remarks are crucial for the consideration of tyre wear in the control modules of automated articulated vehicles, and pave the way for researchers and manufacturers to consider them for the vision of zero-emission transport systems. The current study could be expanded to consider more advanced models, i.e., vehicle and tyre wear models, and a real-time algorithm for trajectory planning. Further work is in progress by the authors to modify the models further and exploit the current contributions.

## Figures and Tables

**Figure 1 sensors-24-03179-f001:**
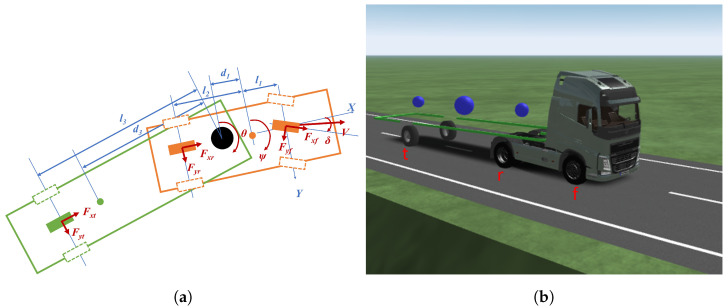
(**a**) Free body diagram of the AVM used in the OCP, and (**b**) the Volvo FH-500 2012 whose parameters are used in the AVM. The tractor has two axles (f + r) and the trailer has one (t).

**Figure 2 sensors-24-03179-f002:**
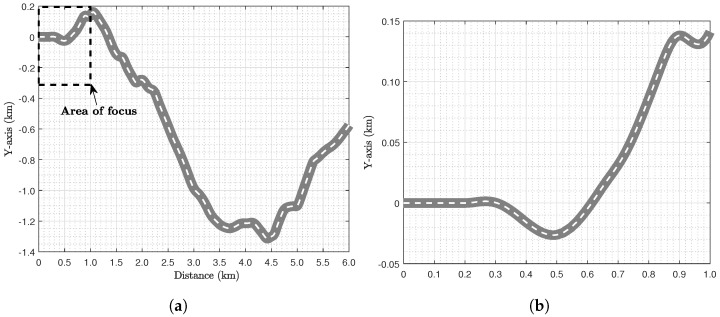
(**a**) Complete road path and (**b**) area of focus for the following analysis.

**Figure 3 sensors-24-03179-f003:**
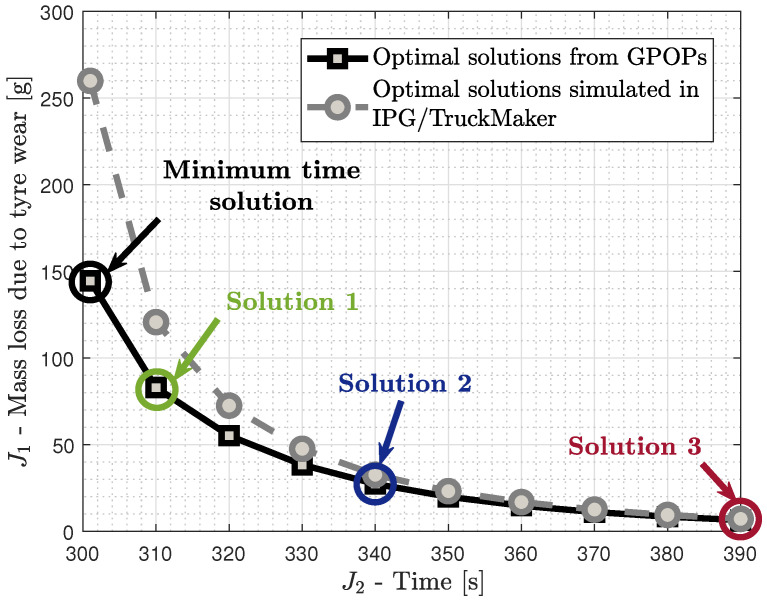
Trade-off between the objectives, i.e., total tyre mass loss ($J_{1}$) and journey time ($J_{2}$), of the optimal solutions obtained from GPOPs, and comparison with the evaluated objectives from the optimal solution simulation in IPG/TruckMaker 13.

**Figure 4 sensors-24-03179-f004:**
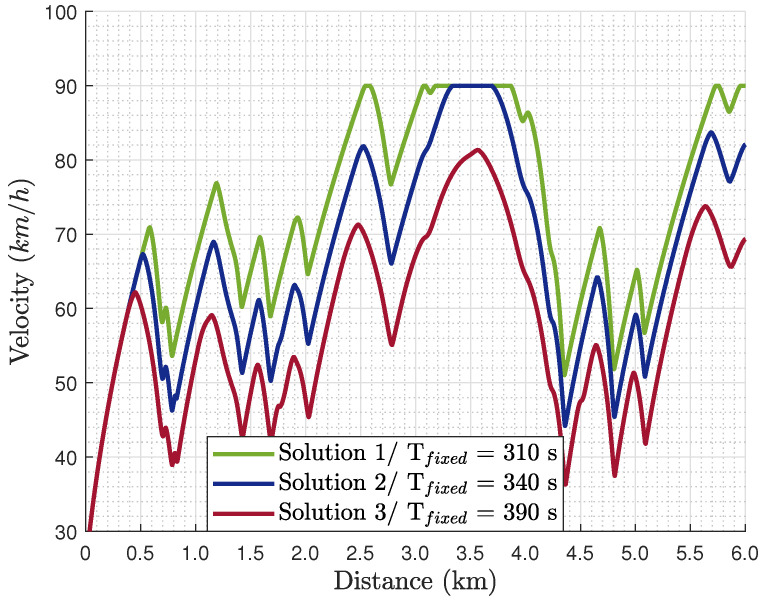
Velocity profile of the three optimal solutions obtained from GPOPs as selected from Figure 3.

**Figure 5 sensors-24-03179-f005:**
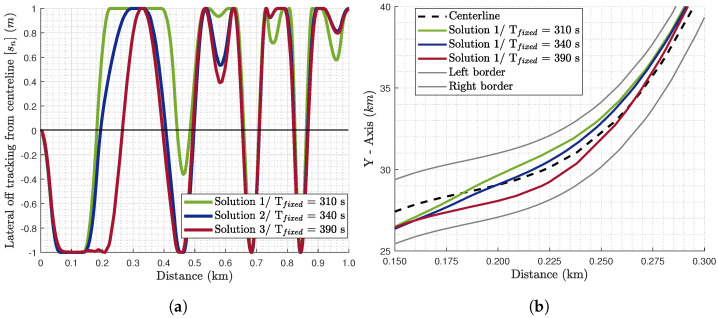
(**a**) The lateral off tracking from the centerline [$s_{n}$] of the three optimal solutions selected in Figure 3. $s_{n}>0$: the vehicle is deviating from the centerline and approaches the left border. $s_{n}<0$: the vehicle approaches the right border. $s_{n}=0$: the vehicle is driving on the centerline. (**b**) The optimal path for these solutions within [0.15 0.30] km for clarity.

**Figure 6 sensors-24-03179-f006:**
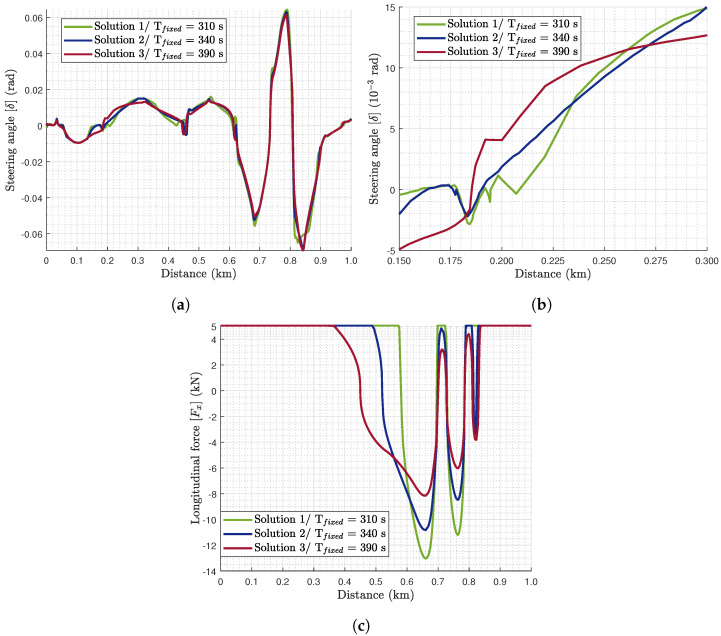
The control inputs of the (**a**) steering angle (δ) for 0–1 km (**b**) and 0.15–0.30 km, and (**c**) the longitudinal force ($F_{x}$) for the three optimal solutions.

**Figure 7 sensors-24-03179-f007:**
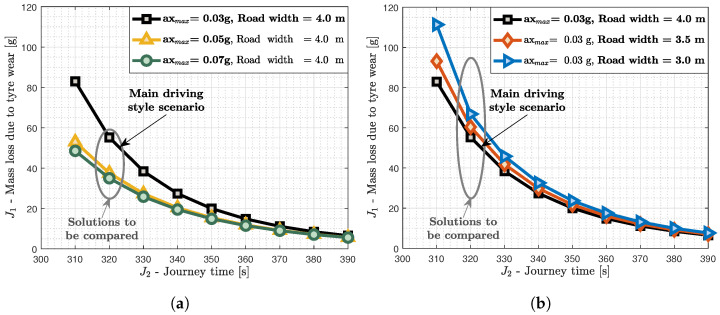
Additional scenarios for investigating the impact of (**a**) vehicle capabilities and (**b**) lateral maneuverability on the trade-off between the tyre mass loss due to wear and the journey time. The main driving style scenario is compared with its corresponding set of alternatives. The three solutions from the vehicle capabilities scenario are studied further.

**Figure 8 sensors-24-03179-f008:**
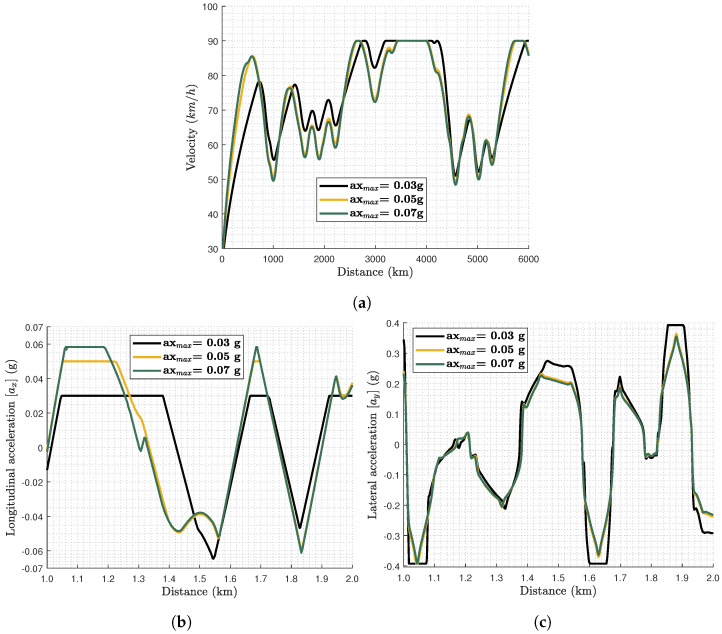
Comparison of (**a**) vehicle velocity ($v_{x}$), (**b**) longitudinal acceleration ($a_{x}$), and (**c**) lateral acceleration ($a_{y}$) for the $T_{fixed}$ case of 320 s obtained by the three different vehicle capability scenarios ($a_{x,max}=$ 0.03 g, 0.05 g, and 0.07 g).

**Table 1 sensors-24-03179-t001:** Comparison of the decrease in tyre mass loss due to wear ($J_{1}$) with the increase in journey time ($J_{2}$) in the optimal solutions from the trajectory planning and the simulations of IPG/TruckMaker 13.

Increase in $J_{2}$	Decrease in $J_{1}$	
	From Solver	From IPG/TruckMaker
From 301 to 310 s	42.2%	53.3%
From 310 to 320 s	33.4%	39.9%
From 320 to 330 s	30.2%	34.4%
From 330 to 340 s	28.9%	31.4%
From 340 to 350 s	27.1%	28.9%
From 350 to 360 s	25.7%	27.2%
From 360 to 370 s	24.8%	25.6%
From 370 to 380 s	23.9%	24.4%
From 380 to 390 s	23.1%	24.1%
**Average Value** =	28.8%	35.3%

**Table 2 sensors-24-03179-t002:** Comparison of the tyre mass loss due to wear ($J_{1}$) from the main driving style scenario in every $T_{fixed}$ case, when the vehicle-capability modifications are applied.

Vehicle Capabilities		
$T_{\mathrm{fixed}}$	Decrease in $J_{1}$	
	${\mathrm{ax}}_{\max}=$ 0.05 g	${\mathrm{ax}}_{\max}=$ 0.07 g
310 s	36.2%	41.5%
320 s	32.1%	36.7%
330 s	28.8%	32.8%
340 s	25.7%	28.9%
350 s	22.7%	25.4%
360 s	20.2%	22.3%
370 s	17.8%	19.5%
380 s	15.5%	16.8%
390 s	13.3%	14.3%
**Average Value** =	23.6%	26.5%

**Table 3 sensors-24-03179-t003:** Comparison of the tyre mass loss due to wear ($J_{1}$) from the main driving style scenario in every $T_{fixed}$ case, when the lateral maneuverability modifications are applied.

Lateral Maneuverability		
$T_{\mathrm{fixed}}$	Increase in $J_{1}$	
	RW= 3.5 m	RW= 3.0 m
310 s	12.3%	34.2%
320 s	9.4%	21.1%
330 s	8.9%	19.3%
340 s	8.9%	19.0%
350 s	8.7%	18.6%
360 s	8.5%	18.3%
370 s	8.4%	18.0%
380 s	8.2%	17.7%
390 s	8.0%	17.5%
**Average Value** =	9.0%	20.4%

## Data Availability

Data are contained within the article.

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
