# Peer review of "Reducing Tyre Wear Emissions of Automated Articulated Vehicles through Trajectory Planningâ€"

_sensors, 2024, doi:10.3390/s24103179_

Round 1
Reviewer 1 Report
Comments and Suggestions for Authors
The structure of the article is correct. However, the manuscript has several shortcomings that should be corrected before publication:
1. the introduction of the article lacks a clearly formulated purpose. The purpose should be clearly formulated and justified.
2. The sketch of the vehicle member is inconsistent with reality (in reality, l2 is larger than d1).
3. In Figure 2b there is no description of the axles.
4. explain and justify whether the formulated objective was achieved.
I ask the authors to answer the following questions:
1. why the simplest model of the component vehicle was chosen for the simulation study - please justify.
2. can the obtained results have practical application?
Author Response
- The introduction of the article lacks a clearly formulated purpose. The purpose should be clearly formulated and justified.
Reply: The authors would like to thank the reviewer for the feedback. The introduction includes a paragraph mentioning the purpose of the paper, but the authors have re-formulate the paragraph to clearly define the purpose and objective of the paper.
- The sketch of the vehicle member is inconsistent with reality (in reality, l2 is larger than d1)
Reply: The authors would like to thank the reviewer for the comment. The figure has been modified.
- In Figure 2b there is no description of the axles.
Reply: The authors have added information about the axles in the figure and the caption.
- Explain and justify whether the formulated objective was achieved
Reply: As mentioned before, this paper focuses on formulating an optimal control problem (OCP) for optimising the trajectory of an articulated vehicle to minimise tyre wear and explore the fundamentals of trajectory planning for minimising tyre wear (preliminary design of the motion planning algorithm, assess the trade-off between journey time and tyre wear, and impact of other aspects on tyre wear minimisation). The research objective is formulated in the introduction through the following research questions:
[i] How could a motion planning algorithm be designed for considering tyre wear minimisation in automated articulated vehicles?
[ii] What is the trade-off between the journey time and the mass loss due to tyre wear?
[iii] To what extent can the vehicle's powertrain and lateral manoeuvrability capabilities affect tyre wear?
Then, in the conclusions, the discussion is summarized and the research questions are answered:
[i] The paper presented a configuration of the optimal control problem for optimising the trajectory of an articulated vehicle to minimise tyre wear significantly.
The minimisation of the tyre wear through the optimal trajectory was validated through simulation of a commercial software.
[ii] The increase of journey time requirements, even to an insignificant amount (3 %), leads to an average decrease of mass loss due to tyre wear around 29 %. This relation is important because it illustrates how a relatively small increase in the journey time leads to such a significant decrease in the mass loss due to tyre wear.
[iii] The employment of articulated vehicles with low powertrain capabilities, leads to greater mass losses due to tyre wear if the journey time requirements are maintained at the same levels with vehicles with greater powertrain capabilities. At the same time, there is no need for excessive increase in powertrain capabilities as after some point the decrease of tyre wear is saturated.
The decrease of the vehicle lateral manoeuvrability leads to the increase of the mass loss due to tyre wear as expected. More specifically, if the road width is decreased by 0.5 m, the mass loss due to tyre wear can be increased by around 10\%.
- Why the simplest model of the component vehicle was chosen for the simulation study - please justify.
Reply: The authors have selected a simplified dynamic model to formulate an efficient optimal control problem while proper constraints were employed to secure that the domain of this approach is within normal driving conditions. The high accuracy and reliability of the model is proven both by experiments and simulations [1,2] regardless its simplicity. The authors also compare the model results with high fidelity models.
[1] Chen, Chieh, and Masayoshi Tomizuka. "Lateral control of commercial heavy vehicles." Vehicle System Dynamics 33.6 (2000): 391-420.
[2] Milani, Sina, et al. "Semitrailer steering control for improved articulated vehicle manoeuvrability and stability." Nonlinear Engineering 8.1 (2019): 568-581.
- Can the obtained results have practical application?
Reply: Regarding the practical application of the current work, the proposed configuration of the optimal control problem could be directly applied in real-time control algorithms with the appropriate considerations, or the extracted paths can be used for references of existing path tracking algorithms. At the same time, the results highlight the importance of considering tyre wear in the control modules, and illustrate that transport or AVs researchers should consider tyre wear in the control modules. Last but not least, through this analysis, critical conclusions for policyholders, researchers and manufacturers can be extracted.
Reviewer 2 Report
Comments and Suggestions for Authors
Reviewer Recommendation
Thank you for inviting me to evaluate the article titled “Trajectory planning of automated articulated vehicles for tyre wear minimisation”. The article proposed the formulation of an optimal control problem for the trajectory optimisation of automated articulated vehicles for tyre wear minimisation. My detailed comments are as follows:
1. The percent match of the manuscript is 31%, the author should revise it down.
2. Nearly half of the abstract is devoted to the current state of research, with insufficient description of the manuscript's methodology, results, and so on.
3. Abbreviations should be indicated in full when they first appear in the manuscript, and used directly in subsequent texts. The full name of AVM in the title of figure 1 has already appeared in the previous text and can be omitted here. The abbreviation COG appears in line 137, yet the full name is not given until line 306. This type of problem needs to be checked in full. Meanwhile, the title of figure 1 lacks a period.
4. The manuscript states only " The equations are presented below as extracted from the literature [20]", and lacks a specific description of equations. 1-7; what does each express?
5. In line 164 of the manuscript, "rotational velocity of a point on the tyre (Rω)", Rω should be the radius of rotation of the tire, not the rotational velocity, please confirm.
6. Please standardize the presentation of numerical values in manuscripts. The numerical expression in lines 178, 181 seems to lack ‘×’.
7. Section 2.3 is a one-paragraph statement, which is relatively small, so please consider whether it would be appropriate as a separate section. Section 3.1 and 3.2 are the same.
8. In Section 4.2, it is mentioned that the articulated vehicle model with higher fidelity is used for evaluation, in which aspects the higher fidelity model is reflected and which aspects are improved.
9. Is it necessary to mark the area of concern in Figure 6 (c)? If so, please box the area of concern more accurately and give the appropriate enlarged picture. The green line of Figure (c) in Figure 6 is annotated as 301s. Please check and modify whether it is correct.
10. The overall beauty of the paper is very important, some of the formulas in this paper choose the right alignment, some choose the center alignment, which needs a unified format. In addition, check whether the subheadings in Figure 6 (b) and Figure 8 (a) have a display problem.
Comments on the Quality of English LanguageMinor editing of English language required.
Author Response
- The percent match of the manuscript is 31%, the author should revise it down.
Reply: This work is an extension of the second author's Master Thesis. Based on our own software, this percentage is mainly due to the similar equations and theory used. However, the authors would be glad to revise the manuscript if the editorial office provides the similarity report and guidelines what is required to change.
- Nearly half of the abstract is devoted to the current state of research, with insufficient description of the manuscript's methodology, results, and so on.
Reply: The abstract has been further improved to incorporate aspects of the methodology and the results. Also, the current state of research has been reduced.
- Abbreviations should be indicated in full when they first appear in the manuscript, and used directly in subsequent texts. The full name of AVM in the title of figure 1 has already appeared in the previous text and can be omitted here. The abbreviation COG appears in line 137, yet the full name is not given until line 306. This type of problem needs to be checked in full. Meanwhile, the title of figure 1 lacks a period.
Reply: The comments have been accommodated.
- The manuscript states only " The equations are presented below as extracted from the literature [20]", and lacks a specific description of equations. 1-7; what does each express?
Reply: The authors have added description about the required equations in the updated manuscript.
- In line 164 of the manuscript, "rotational velocity of a point on the tyre (Rω)", Rω should be the radius of rotation of the tire, not the rotational velocity, please confirm.
The issues has been accommodated. The authors intended to write : ωRω
- Please standardize the presentation of numerical values in manuscripts. The numerical expression in lines 178, 181 seems to lack ‘×’.
Reply: The authors have standardized the presentation. The authors will align with the editorial office if anything else is required by the journal.
- Section 2.3 is a one-paragraph statement, which is relatively small, so please consider whether it would be appropriate as a separate section. Section 3.1 and 3.2 are the same.
Reply: Section 3.1 and 3.2 (currently 3.1) have been combined, but Section 2.3 (currently 2.4) has been maintained as a separate section since it summarizes the limitation of both models (currently Sections 2.1 and 2.3).
- In Section 4.2, it is mentioned that the articulated vehicle model with higher fidelity is used for evaluation, in which aspects the higher fidelity model is reflected and which aspects are improved.
Reply: The simulations with a higher fidelity vehicle model are conducted with IPG TruckMaker. This is highlighted in the manuscript.
IPG TruckMaker simulation solution is specifically tailored to the requirements relevant for the development and testing of heavy-duty vehicles such as trucks, construction vehicles, busses, semi-trucks, heavy-duty semi-trucks and special vehicles. TruckMaker enables the accurate modelling of real-world test scenarios in the virtual world and increases the agility of development processes.
The model used in TruckMaker is the demo model, and considers among others: suspension dynamics, advanced tyre dynamics, individual wheels, engine dynamics, aerodynamics etc.
Some of these limitations are mentioned at Section 2.4.
- Is it necessary to mark the area of concern in Figure 6 (c)? If so, please box the area of concern more accurately and give the appropriate enlarged picture. The green line of Figure (c) in Figure 6 is annotated as 301s. Please check and modify whether it is correct
Reply: The marking of the area of concern has been removed. The legend has been corrected.
- The overall beauty of the paper is very important, some of the formulas in this paper choose the right alignment, some choose the center alignment, which needs a unified format. In addition, check whether the subheadings in Figure 6 (b) and Figure 8 (a) have a display problem.
Reply: The author have modified the alignment of all the equations. Figure 6 and 8 were also modified according to the previous comments. The authors will check the display problem with the editorial office.
Reviewer 3 Report
Comments and Suggestions for Authors
1. In Subsection 2.1.1, it is suggested to give a picture to help the reader understand the derivation of Equations 8 to 11.
2. Line 284: According to the manuscript, the author stated that “Therefore, according to the literature, the steering angle is constrained based on Equation 34.”, Please give the reference for the range of steering angles in Equation 34.
3. Line 287: Please give the basis for setting the steering rate in Equation 35.
4. Line 290 and 291: Please give the basis for setting the hitch angle and the hitch rate in Equations 36 and 37.
5. Line 294: Please give the basis for setting the slip angles in Equation 38.
6. In the paper, the relative change of mass loss due to tyre wear was given, and the reviewer also wants to know the absolute value of mass loss due to tyre wear.
Author Response
- In Subsection 2.1.1, it is suggested to give a picture to help the reader understand the derivation of Equations 8 to 11.
Reply: The authors would like to thank the reviewer for the suggestion. They had omit the figure and added a reference to not overload further the readers.
- Line 284: According to the manuscript, the author stated that “Therefore, according to the literature, the steering angle is constrained based on Equation 34.”, Please give the reference for the range of steering angles in Equation 34.
Reply: The authors have incorporated a reference.
- Line 287: Please give the basis for setting the steering rate in Equation 35.
Reply: The authors have incorporated a reference.
- Line 290 and 291: Please give the basis for setting the hitch angle and the hitch rate in Equations 36 and 37.
- Line 294: Please give the basis for setting the slip angles in Equation 38.
Reply: The authors have used these values based on offline simulation with commercial software. The authors have included also references that have used for validating the bounds.
[1] Kabbani, Tarek, et al. "Model Validation of Articulated Heavy-Duty Vehicle via IPG for Tracking Controller Design." Transportation Research Procedia 72 (2023): 681-687.
[2] Oreh, SH Tabatabaei, R. Kazemi, and S. Azadi. "A new desired articulation angle for directional control of articulated vehicles." Proceedings of the Institution of Mechanical Engineers, Part K: Journal of multi-body dynamics 226.4 (2012): 298-314.
- In the paper, the relative change of mass loss due to tyre wear was given, and the reviewer also wants to know the absolute value of mass loss due to tyre wear.
Reply: The tables illustrate the relative changes. The absolute value of mass loss for all the scenarios is illustrated in Fig. 3 and 7. The range of the absolute value for all the tyres (i.e., six tyres) is 5 to approximately 145 gr, with the maximum value to be for the minimum time solution.
Reviewer 4 Report
Comments and Suggestions for Authors
Nice paper, good English, easy to read, and interesting. I have only minor comments/suggestions:
1. Paper title should be precise and informative - it is very good.
2. Abstract should be a compressed version of the paper, written in easy language, understandable to a very wide audience, free of statements or forms which distract from the narrative - your abstract is almost good. The only two things that are not ok for me are: (1) the very first sentence, while not all researchers consider the rubber dust to be microplastic, so you should not make such an inclusive statement, (2) the phrase "demand for increase in journey time" seems odd to me, especialy the word "demand".
The introduction is good, the equations seem to be good, figures are informative, although I don't like the legend of Fig.7. - it should be sorted by ax and then rw and it is a mess now.
The last thing is the term "mass loss" you never say "what object" is weighted here, but the term "mass loss due to tyre wear" suggests that it is the car or truck. A truck has tires. A car also has tires. But in the tables (and in text) you say that you get "20.4%" of "percentage of increase of the decrease of mass" ((did you mean that the truck is 20% lighter now!?)). Or Tab.2. 26% lighter? It took me a lot of time to understand, that the "percentage of decrease in J1" (row 2) does NOT describe the columns, but only the row 3, and that the columns are being decribed by the row 1. Don't do that. Do not mix two concepts in the table header. I think the reader does not have time to decipher what was your intention. I think that the purpose of a table is to show numbers to make it easier for the reader to understand, not harder.
Besides these issues, I think the paper is nice and may be useful.
99. It is considered to be rude and bold to include numerous reference items of your own authorship. It is either bold or justification of auto-plagiarism. One or two items might be o.k. but not 5.
Author Response
- Nice paper, good English, easy to read, and interesting. I have only minor comments/suggestions:
Reply: The authors would like to thank the reviewer for the feedback.
- Paper title should be precise and informative - it is very good.
Reply: The authors would like to thank the reviewer for the feedback.
- Abstract should be a compressed version of the paper, written in easy language, understandable to a very wide audience, free of statements or forms which distract from the narrative - your abstract is almost good. The only two things that are not ok for me are: (1) the very first sentence, while not all researchers consider the rubber dust to be microplastic, so you should not make such an inclusive statement, (2) the phrase "demand for increase in journey time" seems odd to me, especialy the word "demand".
Reply: The authors have accommodated the comments by the reviewer. The first sentence was replaced, while the phrase "demand for" has been omitted. This has been accommodated across the whole manuscript.
- The introduction is good, the equations seem to be good, figures are informative, although I don't like the legend of Fig.7. - it should be sorted by ax and then rw and it is a mess now.
Reply: The authors have split the material of Fig. 7 in two figures for better clarity according to the reviewers comments.
- The last thing is the term "mass loss" you never say "what object" is weighted here, but the term "mass loss due to tyre wear" suggests that it is the car or truck. A truck has tires. A car also has tires. But in the tables (and in text) you say that you get "20.4%" of "percentage of increase of the decrease of mass" ((did you mean that the truck is 20% lighter now!?)). Or Tab.2. 26% lighter? It took me a lot of time to understand, that the "percentage of decrease in J1" (row 2) does NOT describe the columns, but only the row 3, and that the columns are being described by the row 1. Don't do that. Do not mix two concepts in the table header. I think the reader does not have time to decipher what was your intention. I think that the purpose of a table is to show numbers to make it easier for the reader to understand, not harder.
Reply: The authors would like to thank the reviewer for highlighting this issue. The mass loss refers to the tyre mass loss due to wear. The authors have reformulated the writing to accommodate this issue. They refer now to the "tyre mass loss due to wear”. Meanwhile, the phrase "percentage of increase" has been modified to "increase". This means that the mass loss has been increased by X % if time increases accordingly.
- Besides these issues, I think the paper is nice and may be useful.
It is considered to be rude and bold to include numerous reference items of your own authorship. It is either bold or justification of auto-plagiarism. One or two items might be o.k. but not 5.
Reply: The authors would like to thank the reviewer for the feedback. Their intention was not to be rude neither justify self-plagiarism. The authors have been among the first groups to investigate vehicle dynamics related countermeasures to reduce tyre wear. Thus, the last years, they developed different approaches (suspension optimisation, active suspensions, motion planning etc.). This intensive effort has led to multiple publications, which are cited in this paper accordingly. However, some more preliminary works have been removed.
Round 2
Reviewer 2 Report
Comments and Suggestions for Authors
The author has carefully revised the manuscript based on the reviewers' comments, and I think the manuscript can be accepted in its current form.
Reviewer 3 Report
Comments and Suggestions for Authors
Authors have answered all my questions and addressed all my concerns in the revision. The revised manuscript is improved in many aspects. I think this paper can be accepted for publication.